# The Road to Generalizable Neuro-Symbolic Learning Should be Paved with Foundation Models

## Abstract

Neuro-symbolic learning was proposed to address challenges with training neural networks for complex reasoning tasks with the added benefits of interpretability, reliability, and efficiency. Neuro-symbolic learning methods traditionally train neural models in conjunction with symbolic programs but they face significant challenges that limit them to simplistic problems. On the other hand, purely-neural foundation models now reach state-of-the-art performance through prompting rather than training, but they are often unreliable and lack interpretability. Supplementing foundation models with symbolic programs, which we call neuro-symbolic prompting, provides a way to use these models for complex reasoning tasks. Doing so raises the question: What role does specialized model training as part of neuro-symbolic have in the age of foundation models? To explore this question, we highlight three pitfalls of traditional neuro-symbolic learning with respect to the compute, data, and programs leading to generalization problems. This position paper argues that foundation models enable generalizable neuro-symbolic solutions, offering a path towards achieving the original goals of neuro-symbolic learning without the downsides of training from scratch.

## 1 Introduction

Foundation models pre-trained on general internet-scale data are now ubiquitous, bringing the benefits of deep learning to downstream applications across several domains [1]. This is achieved primarily via prompting techniques as well as finetuning for more niche use cases. Their success is driven by scaling up both the training data and model parameters, leading to predictable performance improvements [2]. Even so, limitations on problems requiring complex reasoning and reliability remain [3, 4]. Further, these systems are fundamentally black-box and lack features like interpretability, which is vital for safety-critical domains such as medicine [5, 6], autonomous driving [7], and aviation [8], and their unpredictable nature raises safety concerns for their real-world deployment [9].

Neuro-symbolic learning is a paradigm that moves towards a solution to these limitations by training deep neural models in conjunction with symbolic reasoning [10]. Figure 1 on the left shows a setup common for neuro-symbolic systems such as Scallop [11], ISED [12], and NeurASP [13]. Here, the task is split into two common subtasks, perception and reasoning. Perception tasks convert raw inputs (text, images, video, etc.) into symbols using deep neural models and the reasoning task uses a symbolic program (e.g. a Python function) to process the symbols from perception [14]. Training the system end-to-end provides several benefits over traditional neural networks, including data efficiency (using smaller datasets and less supervision), generalizability, and interpretability (the intermediate symbols can be examined and the program offers a faithful explanation of the output). Despite these benefits, neuro-symbolic training is often impractical due to scalability challenges, which are significantly rooted in the typical absence of direct supervision for the intermediate symbols produced by the neural model before they are processed by the symbolic reasoner [15].

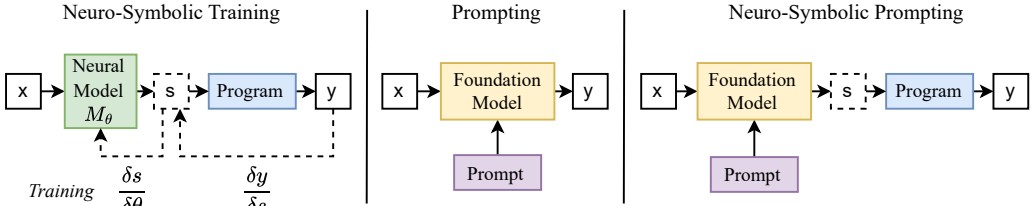

Figure 1: Neuro-symbolic training consists of neural models whose results are fed into a program to produce the desired output (shown on the left). Training the neural models in neuro-symbolic training is a significant challenge due to not having supervision for the intermediate symbols, denoted as $s$. Shown in the middle is the modern foundation model prompting paradigm where a user provides a prompt and their input to a foundation model which then produces the output. On the right, we show the foundation model approach to neuro-symbolic which like *prompting* does not require training, but uses a symbolic component like neuro-symbolic *training*.

While a major challenge in traditional neuro-symbolic training is learning the neural component, foundation models can now perform many tasks requiring strong input understanding without any additional training [1, 16, 17]. The common prompting setup is shown in the middle of Figure 1 where there is now just a prompt rather than the training and program. As foundation models are trained on such large amounts of data, prompting can even offer better performance and robustness than a training-based method using less data [1]. This raises the following question: *What role does specialized model training as part of neuro-symbolic have in the age of foundation models?*

In this paper, we argue that **building neuro-symbolic atop frontier foundation models allows for achieving the benefits of neuro-symbolic learning—program reliability and symbol interpretability—without the disadvantages that come with training.** We term the replacement of neural components in the traditional neuro-symbolic training setup with foundation models as *Neuro-Symbolic Prompting*, and show its setup in Figure 1 on the right. Neuro-symbolic prompting uses prompted foundation models to perform the perception task of extracting symbols, while a symbolic program (e.g. a Python program) is used to reason over the detected symbols. Several prior works fall into the category of neuro-symbolic prompting including Vieira [18] which combines foundation models with Scallop, SatLM [19] which uses LLMs to convert input problems into constraint solver input, and LLMFP [20] which solves planning problems by formalizing them as constrained optimization problems which are solved by a discrete optimizer.

Our experiments uncover several pitfalls of traditional neuro-symbolic training including unnecessarily training models when prompting is now available, overfitting to labeled datasets, and reliance on a single program to provide learning signal for the correct behaviors. On the other hand, we show that neuro-symbolic prompting provides opportunities for enabling reliability and interpretability of foundation models without the pitfalls associated with training in neuro-symbolic training. In light of these findings, we look towards future research on neuro-symbolic prompting, where we highlight the problem of autonomously inferring the symbols and program as the significant remaining frontier.

## 2   Pitfalls

The neuro-symbolic training paradigm enables the use of explicit programs in an end-to-end differentiable manner, which allows for training models for reasoning tasks using less supervision, data, and compute compared to traditional deep learning techniques. However, in the age of foundation models, a well-crafted prompt often performs just as well or better than training a deep neural network from scratch. neuro-symbolic prompting techniques, which build neuro-symbolic systems on top of prompted foundation models, provide many of the benefits of neuro-symbolic training, with the additional generalization benefits of foundation models, without the need for training of specialized models from scratch. We thus observe that the neuro-symbolic training paradigm faces three main pitfalls in the age of foundation models: the *compute pitfall*, the *data pitfall*, and the *program pitfall*. Our quantitative results which describe each pitfall are shown in Table 1 which we will describe in detail with further evidence in the following sections.

Table 1: Results for the pitfalls across five datasets (D1-D4). D1: Sum5, D2: HWF5, D3: CLUTRR, D4: Leaf, D5: CLEVR. For symbol hallucination, we measure the fraction of times that the neuro-symbolic training method produces the true intermediate symbols, specifically among cases when the neuro-symbolic training method's final answer is correct but the neuro-symbolic prompting method's final answer is wrong. A value of 1 means the performance gap between neuro-symbolic training and neuro-symbolic prompting is due to inferring the true symbols while a low value means the correct answer is reached via hallucinated symbols. The '*' indicates a human evaluation.

| Pitfall Cat. | Aspect / Condition | Metric | D1 | D2 | D3 | D4 | D5 |
|---|---|---|---|---|---|---|---|
| **Compute Pitfall** | Neuro-Symbolic Training vs. Prompting Perf. | Acc. Diff. ($NS_{train} - NS_{prompt}$) | +0.11 | +0.25 | -0.39 | +0.31 | -0.15 |
| **Data Pitfall** | Robustness at 3% Noise | Acc. Drop ($NS_{train}$) | -0.02 | -0.21 | – | -0.11 | -0.19 |
| | | Acc. Drop ($NS_{prompt}$) | -0.03 | -0.06 | – | -0.06 | -0.07 |
| **Program Pitfall** | Symbol Hallucination | Symbol GT Match | – | – | 0.00 | 0.16* | 0.14 |

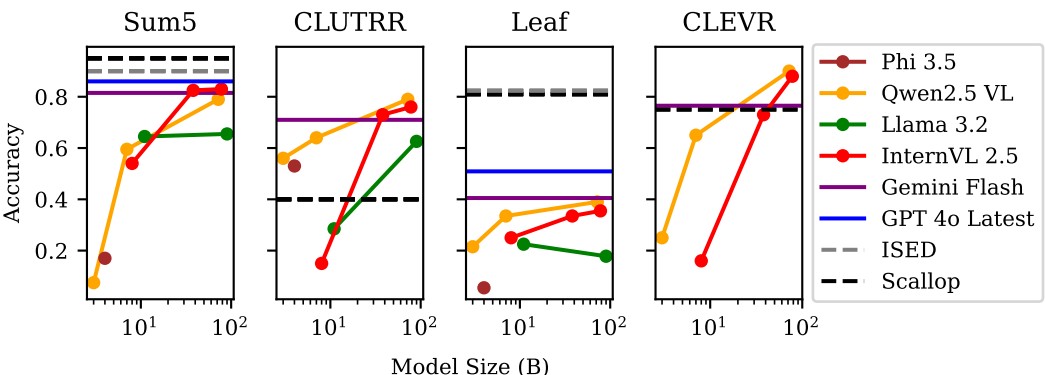

Figure 2: Performance of neuro-symbolic prompting for four benchmarks as model size increases compared to Scallop and ISED, two baseline neuro-symbolic training methods. As model size increases, the gap between neuro-symbolic training and neuro-symbolic prompting increasingly vanishes. There is still a considerable gap for the Leaf dataset which we discuss with regard to *the program pitfall* in Section 2.3. See Table B.1 for full results on all five datasets.

## 2.1 The Compute Pitfall

From its inception, neuro-symbolic training was proposed as a solution for tasks requiring both deep learning and complex reasoning where, formerly, knowledge that was otherwise human-specifiable would instead need to be learned indirectly by extensive training of large neural networks from scratch [21]. While the neuro-symbolic training paradigm relies on training, the models trained can be smaller and more specialized, learning concepts with less data and less overall compute.

The proliferation of foundation models changes the underlying assumptions of this paradigm. If one can replace the models from traditional neuro-symbolic training with foundation models that no longer require additional training, they may forgo the compute traditionally required in neuro-symbolic training. Therefore, how beneficial is the training component of neuro-symbolic training?

To answer this question, we start by comparing two neuro-symbolic training techniques, Scallop [11] and ISED [12], against neuro-symbolic prompting instantiated with various open and proprietary foundation models. For foundation models, we consider recent multimodal LLMs of a diverse set of sizes and capabilities. We evaluate all methods on five benchmarks frequently used in prior work on neuro-symbolic training methods. As benchmarks, we consider the Sum5 dataset [11] which asks for the sum of five MNIST [22] handwritten digits, the HWF5 dataset [23] which asks for the result of evaluating a handwritten arithmetic expression, the CLUTRR dataset [24] which asks for the relationship between people described in text, the Leaf dataset [12, 25] which asks for a plant's name from an image of a leaf, and finally the CLEVR dataset [26] which asks questions about an image containing various objects. For foundation models, we use the Llama 3.2 [27], Qwen 2.5 VL

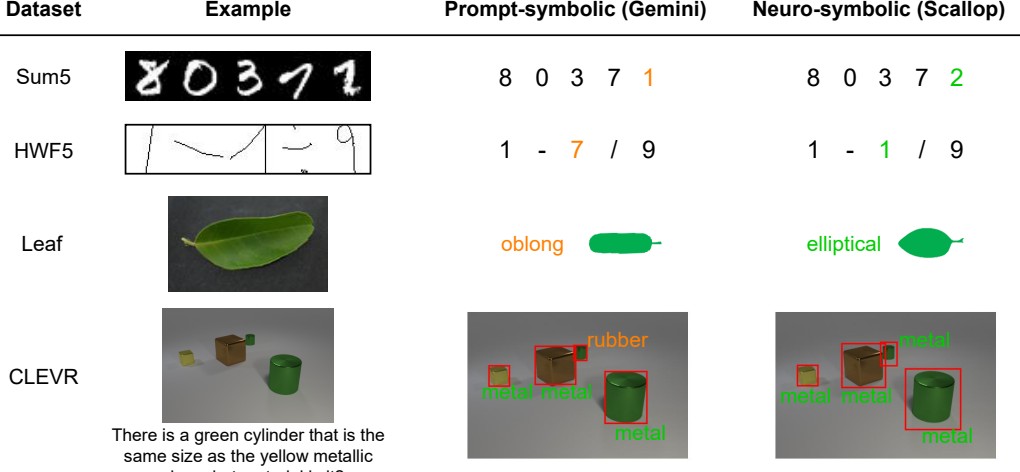

| Dataset | Example | Prompt-symbolic (Gemini) | Neuro-symbolic (Scallop) |
|---|---|---|---|

Figure 3: Examples of *the data pitfall* from four benchmarks showing cases where the neuro-symbolic prompting method using Gemini makes an "error". In contrast, the neuro-symbolic training method is "correct." These predictions, marked as errors from neuro-symbolic prompting, reflect cases where some symbols are ambiguous or hard to determine from the input. In contrast, the neuro-symbolic training method appears to have memorized noise and biases in the dataset to get the "correct" symbols. For example, for the Leaf dataset, the leaf is folded so that it looks oblong (which is predicted by Gemini), but Scallop predicts elliptical, which is correct based on this species of leaf.

[28], InternVL 2.5 [29], and Phi 3.5 [30] family of open models along with Gemini 2.0 Flash [31] and GPT 4o [32]. See Appendix B for further information on our experimental setup.

The first row of Table 1 shows the accuracy difference between the best neuro-symbolic training and the best neuro-symbolic prompting method for all five datasets, and Figure 2 shows a snapshot of four benchmarks with increasing model size. In each subplot of Figure 2, Scallop's performance is denoted by a black dashed line, whereas the solid lines indicate the performance of neuro-symbolic prompting. We plot the performance of foundation models against their size (number of parameters). In all cases, the foundation models are strictly prompted, without any training or fine-tuning.

Consider the graphs for the CLUTRR and CLEVR benchmarks. In both cases, we can see that the smallest versions of the foundation models perform worse than Scallop, where the neural models are specifically trained for that particular task. However, notice that as the size of the foundation models increases, they progressively close the performance gap, with their largest versions eventually outperforming Scallop's trained models. In the case of both Sum5 and Leaf, the largest foundation models are unable to outperform Scallop or ISED, with the performance gap being more significant for Leaf. However, in both cases, the gap still narrows with scale, and we will address reasons why the gap remains large for these datasets in the following sections on the data and program pitfalls.

These results show foundation models can generally convert raw input to symbols, nearing the performance of specialized, task-specific models. Further, as the size of foundation models increases, they can replace, or come closer to matching, task-specific models trained through neuro-symbolic training techniques. This encapsulates our first pitfall: *spending compute on training specialized perception models in neuro-symbolic learning has diminishing returns as the performance gap with neuro-symbolic prompting shrinks with scale.* Naturally, we want to understand why there may be performance gaps for the Sum5 and Leaf benchmarks; closer examination of model behaviors over these benchmarks reveals the next two pitfalls.

## 2.2 The Data Pitfall

The performance gap often arises when neuro-symbolic prompting errs on ambiguous symbols, while traditional neuro-symbolic trained models predict dataset-conformant "correct" symbols based on data biases or noise. This finding sheds light on the data pitfall: *neuro-symbolic training with*

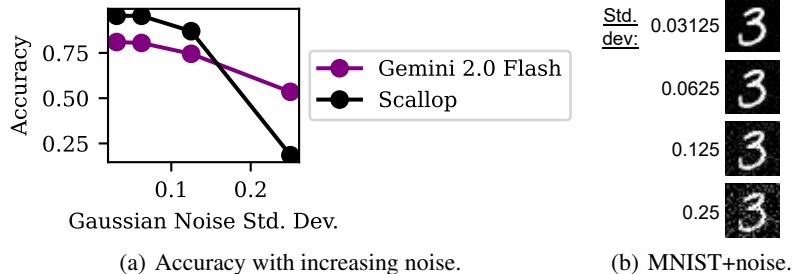

(a) Accuracy with increasing noise.    (b) MNIST+noise.

Figure 4: Neuro-symbolic *trained* neural models memorize dataset biases rather than learn general concepts like foundation models.

*specialized datasets, as opposed to large-scale foundation model pretraining, encourages overfitting to dataset particularities.*

We quantify overfitting from neuro-symbolic training by studying generalization under slight distribution shifts. The second row of Table 1 shows the performance drop after adding 3% Gaussian noise to the four image datasets for neuro-symbolic prompting is significantly lower than for neuro-symbolic training, highlighting the greater generalizability from foundation models. In Figure 4, we show neuro-symbolic training and neuro-symbolic prompting performance on the Sum5 task with increasing noise levels. The noisy data shown in Figure 4(b) is small enough to not affect the image visibility. Despite this, the addition of noise significantly drops neuro-symbolic training performance below that of neuro-symbolic prompting.

In Figure 3, we show several qualitative examples which highlight the data pitfall in terms of cases where the data itself is ambiguous, yet Scallop makes the correct prediction. We include further examples in Figure D.2 and Figure D.3. For instance, consider the MNIST digits used in one sample from the Sum5 dataset. While the first four digits are relatively clear, the last digit is ambiguous. Gemini predicts it to be a "1", while Scallop predicts it as a "2". Even though the digit appears to be closer to a "1", the correct answer is "2", so Scallop infers the correct sum. We attribute this discrepancy to the possibility that the model trained by Scallop has memorized this particular image to be a "2", while Gemini's prediction is not biased by the peculiarities of the MNIST dataset. We see similar behavior in other benchmarks including HWF5, Leaf, and CLEVR shown in Figure 3.

## 2.3 The Program Pitfall

The neuro-symbolic training paradigm assumes the reasoning program is provided by domain experts and relies on it as a form of supervision. As such, the concepts learned by the perception models are highly dependent on the program itself. Since this supervision is relatively weak, with ground truth not available for the perception subtasks, the neural network may learn to hallucinate, or mispredict, symbols that still result in the correct answer due to the reasoning program. This results in the program pitfall: *using programs as a component in neuro-symbolic training can lead to the neural component hallucinating symbols*.

Figure 5 shows examples of the program pitfall in three datasets, and we provide additional examples in Figure D.4. Consider the Leaf dataset, where there is still a large gap between neuro-symbolic training and neuro-symbolic prompting performance shown in the first row of Table 1. The reasoning program in this case is a decision tree over the edge, shape, and texture of the leaf. This program is specified in a forestry database [33] developed for identifying leaves. Here, we find that many of the neuro-symbolic prompting errors correspond to cases where these features are extremely challenging to identify simply from a given image of a leaf.

For example, for the leaf shown in Figure 5, Gemini identifies that it has an "entire" margin, meaning it is smooth, while Scallop identifies it as "serrulate," meaning it has a finely serrated edge. From just the image, it is difficult to determine whether the margin is serrulate or entire, even though identifying this is vital to reaching the correct answer via the program. Even more ambiguous is the texture of the leaf, which Scallop predicts as smooth (resulting in the correct final classification) while Gemini predicts as glossy even though the texture is not clear from the image.

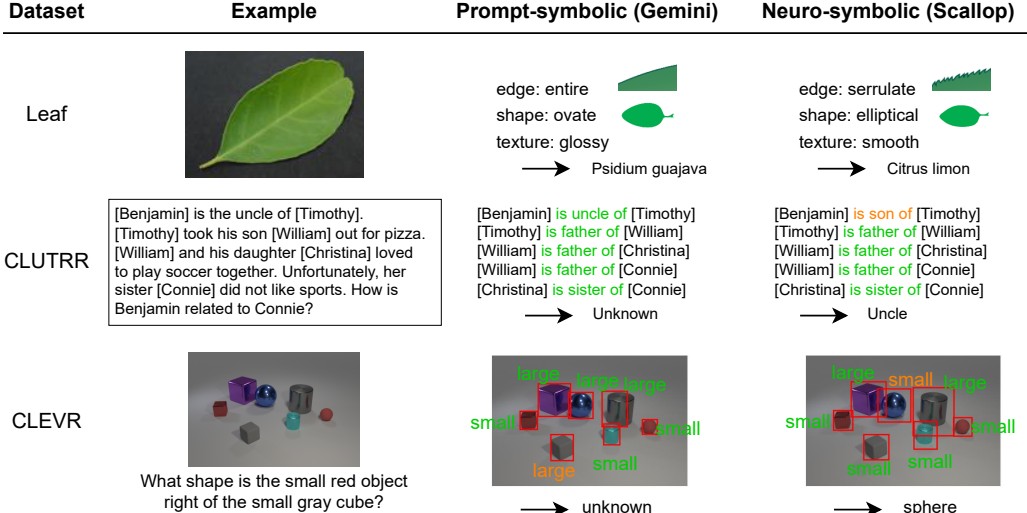

Figure 5: Examples of *the program pitfall*. All examples reflect errors made by neuro-symbolic prompting where the neuro-symbolic training method (Scallop) produced the correct answer. We see that the neuro-symbolic training method reaches the correct answer for the wrong reasons or even identifies seemingly undetectable symbols to reach the correct answer. For CLUTRR, the neuro-symbolic prompting method extracts the correct symbols, but the symbolic program cannot deduce an answer, while the neuro-symbolic training method hallucinates incorrect symbols and reaches the correct answer. For CLEVR, the neuro-symbolic training method incorrectly identifies a blue sphere as "small" when it appears large while neuro-symbolic prompting identifies the blue sphere as large but misjudges a cube as large which was vital to the question.

163 To validate that these concepts are not immediately apparent from the image, we perform a human
164 evaluation using Prolific [34] to determine if the outputs of Scallop's trained neural model are really
165 correct for the cases where neuro-symbolic prompting with Gemini gets the wrong answer. Results
166 are shown in Table 2 where we see that a majority of respondents disagree with Scallop's neural
167 outputs for margin and shape classification even though they result in the correct answer. For texture
168 classification, the inter-annotator agreement is so low that it indicates people cannot reliably determine
169 leaf texture from the images. Full details of this human evaluation are in Appendix C.

Table 2: Human evaluation of foundation model errors on the Leaf classification dataset. We compare foundation model predictions leading to the wrong classification with the Scallop predictions which produce the correct answer. We find that Scallop leads to "symbol hallucination" since humans overall disagree with Scallop predictions even if they lead to the right answer.

| Symbol Category | % Scallop Wrong | Agreement (Cohen's Kappa) |
| --- | --- | --- |
| Margin | 58.8 | 0.32 |
| Shape | 60.0 | 0.34 |
| Texture | 46.7 | 0.05 |

170 We call this behavior *symbol hallucination*, since the neural model in neuro-symbolic training
171 identifies symbols which do not appear present in the input, but their identification leads to the
172 correct output from the program. A similar phenomenon called *reasoning shortcuts* is identified in
173 prior work [35] where reasoning shortcuts occur from a program with multiple choices of symbols
174 which result in the correct answer. We use the term symbol hallucination to also include the case
175 where the program is not perfect, so the wrong symbols are necessary to reach the correct answer.
176 We also see a similar behavior on the CLUTRR dataset in Figure 5 where, due to limitations of
177 the reasoning program, neuro-symbolic prompting gets the wrong answer even though it correctly
178 identifies the symbols, while Scallop hallucinates the incorrect fact that "Benjamin is the son of
179 Timothy," resulting in a correct answer that was spuriously derived. For the CLEVR example, both

Scallop's model and neuro-symbolic prompting mispredict some symbols due to misjudging object size, but the neuro-symbolic training method still results in the correct answer since the program happens to ignore the mispredicted object. In this case, neuro-symbolic training gets the right answer, but has not actually learned the desired distinction between small and large objects.

Focusing on the problems which Scallop answers correctly but neuro-symbolic prompting with Gemini answers incorrectly, we report the fraction of solutions using the correct intermediate symbols in Table 1, when symbol annotations are available. This shows that overwhelmingly, the performance advantage of Scallop over neuro-symbolic prompting does not come from more "correct" intermediate symbol prediction, but rather from hallucinating symbols which happen to reach the correct answer.

# 3    Opportunities

Given these pitfalls, what is neuro-symbolic's merit today? We argue its core principles offer key opportunities with foundation models that enable general input perception which can benefit from the rigor of symbolic programs.

## 3.1    Program Reliability

Compared to pure foundation model prompting, using a symbolic program in a neuro-symbolic training or neuro-symbolic prompting approach can improve accuracy and provide reliability. Recent results have shown that combining foundation models with explicit programs in various neuro-symbolic prompting configurations improves performance for mathematical reasoning tasks while providing reliability and trustworthiness that were lacking through pure prompting [36–38]. Beyond mathematical reasoning, a neuro-symbolic prompting approach is beneficial for any task involving symbolic computation, since performing exact symbolic computation will always be more accurate and reliable than a neural approximation.

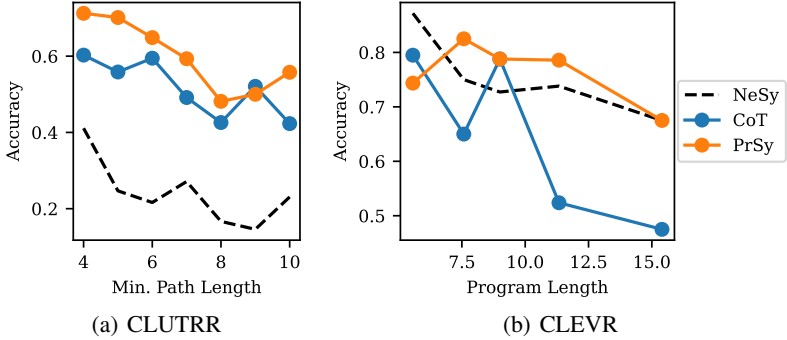

(a) CLUTRR          (b) CLEVR

Figure 6: Performance of end-to-end prompting of Gemini-2.0-Flash compared to neuro-symbolic prompting (Gemini-2.0-Flash with a program) on CLUTRR and CLEVR examples of increasing complexity. Complexity for CLUTRR is the minimum number of reasoning steps needed, and complexity for CLEVR is the length of the program required to answer a sample's question.

As an example, we take the CLUTRR benchmark which asks about the relationship between two people described in a paragraph and compare the accuracy of neuro-symbolic prompting to pure prompting (using Chain of Thought [39]). The results shown in Figure 6(a) demonstrate that neuro-symbolic prompting achieves consistently high accuracy with increasing question complexity while prompting lags in performance. As such, prompting serves as an approximation for symbolic behavior, but using a real symbolic program via neuro-symbolic prompting yields higher accuracy. Similar behavior is shown for the CLEVR dataset in Figure 6(b) where neuro-symbolic prompting results in higher and more stable performance than prompting with Chain of Thought.

## 3.2    Symbol Interpretability

In addition to the benefits from using program execution rather than a neural approximation, the other significant benefit is that the symbol extraction step provides a means for interpretability,

which was a major motivation for the emergence of neuro-symbolic training [40]. As foundation models become more prevalent in real-world applications, this need only grows. These paradigms are not orthogonal in this manner; foundation models can offer additional interpretability compared to traditional neuro-symbolic training since large-scale pretraining is less likely to overfit to artifacts of any one dataset [41].

An example of how intermediate symbols are useful for interpretability can be seen in the CLEVR example in Figure 3. In this case, neuro-symbolic prompting results in the wrong answer of "rubber." The nature of neuro-symbolic prompting allows us to debug why the model produced the wrong answer by investigating the intermediate symbols input to and resulting from the reasoning program. In this case, we see that the green cylinder was misidentified as rubber, since it is hard to tell the cylinder's material. However, a prediction from pure prompting would have been difficult to debug and understand due to a lack of interpretable intermediate symbols and any guarantees that Chain of Thought based explanations are faithful to themselves, meaning they explain the true mechanism behind determining the final answer.

# 4   Looking Ahead

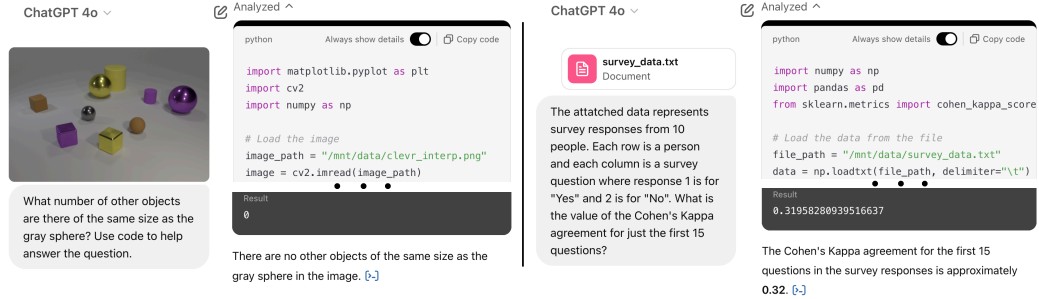

Figure 7: Examples of ChatGPT code execution. Generated code is executed for solving the problem, but we can see in the example on the left from CLEVR that the model can produce the wrong code. For the example on the right, the model correctly calculates the Cohen's Kappa of the attached data.

As foundation models continue to scale, the neuro-symbolic training pitfalls will only become more apparent, and the opportunities more important. As we demonstrate in this paper, foundation models are now highly capable for general input processing/understanding tasks which neural models were traditionally trained for in neuro-symbolic training. The remaining problem in neuro-symbolic training is no longer learning to identify symbols, but *determining which symbols and what program to use for a problem.*

We empirically demonstrate in Section 3.2 that the use of programs in a neuro-symbolic prompting setup offers the potential for reliable and accurate symbolic reasoning, but the use of human specified programs greatly limits the practical applicability and performance of the method. As such, effectively synthesizing programs over foundation model symbols is still an open problem.

There is now growing interest in this problem with methods that mostly focus on prompting or finetuning foundation models for generating the program in a neuro-symbolic prompting setup [36–38, 42]. Industry has also taken interest as shown with the release of OpenAI's Code Interpreter [43] and Google's Gemini code execution [31] which can write and execute its generated code as well as OpenAI's Operator [44] which writes code for performing actions on a computer. As shown in the example on the right of Figure 7, currently available neuro-symbolic prompting tools are already useful for relatively simple data analysis tasks. However, for more complicated tasks such as CLEVR questions, the example on the left shows these methods are ineffective, potentially resulting in even worse performance than pure prompting.

# 5   Related Work

**Neuro-Symbolic Training Methods** For a survey on neuro-symbolic training methods see Garcez et al. [40]. Neuro-symbolic training methods often consist of a neural perception component followed

by symbolic reasoning [14, 45]. There are now several frameworks for constructing such neuro-symbolic training setups including DeepProbLog [46], Scallop [11], NeurASP [13], LTN [47], ISED [12], and Dolphin [48]. All these approaches make various assumptions regarding program differentiability, and provide different levels of scalability. There are also neuro-symbolic methods focusing on learning logic rules, such as NLIL [49] and DRUM [50], but these are not the central focus of our comparison, which centers on systems with explicit, often pre-defined, symbolic programs for reasoning over neurally perceived symbols.

**Neuro-Symbolic Prompting Methods** Existing work which incorporates foundation models with neuro-symbolic training often uses a foundation model to generate code which is then executed [36]. These approaches either use prompting to produce explicit code [18, 20, 36–38] or finetuning for code generation [42, 51]. There is also work on directly finetuning foundation models for symbol extraction [52]. Finally, neuro-symbolic training has also been combined with foundation models to help design new neuro-symbolic training datasets [23] and to develop datasets for finetuning of foundation models [53].

**Challenges in Neuro-Symbolic** Several works have recently identified challenges and misconceptions with the common neuro-symbolic training setup. Reasoning shortcuts, first identified by Marconato et al. [54], are cases where a neuro-symbolic training method learns symbols with the wrong semantics, leading to poor performance on programs using the same symbols in different ways. Reasoning shortcuts have been further studied in neuro-symbolic training settings [35, 55, 56] as well as other research in machine learning more broadly [57, 58]. Reasoning shortcuts are a consequence of the program pitfall. Another challenge comes from the common assumption on independence of all symbols, which often does not hold [59]. Similarly, it is assumed that the detected symbols should display locality, or being influenced by a subset of input features [60]. As observed by Raman et al. [60], training in a neuro-symbolic training setup actually does not result in symbols with the desired locality, another instance of the program pitfall.

# 6  Alternate Views

While we focused on *prompting* of foundation models to argue for their importance in generalizable neuro-symbolic learning, we also address several alternate views. First, finetuning foundation models reintroduces training, risking Section 2's pitfalls, but starting from a capable base model can minimize generality degradation. We therefore see finetuning of foundation models in a neuro-symbolic learning system as a middle ground between neuro-symbolic training and neuro-symbolic prompting that in many cases will be preferable to prompting, however we focus on prompting in this paper as it is already enough to support our argument and finetuning becomes less necessary as foundation models become more capable [61]. Second, training may still be *necessary* for specialized domains lacking pretraining data. In this case where prompting will be ineffective, finetuning should be used to learn effectively from the minimal amount of data available. This is because finetuning foundation models is much more data efficient than training from scratch [62]. Finally, one can argue that the use of small specialized models in neuro-symbolic, while potentially performance matched by larger foundation models, is necessary for resource constrained scenarios. Similar to finetuning, we see merit in future methods which distill or compress foundation models into specialized models in a neuro-symbolic system, trading some generalizability for lower resource demand.

# 7  Conclusion

In this paper, we took a critical look at traditional neuro-symbolic training in the age of foundation models. Neuro-symbolic training, as originally proposed, was meant to address the limitations of deep learning on complex reasoning problems, as well as its lack of reliability and interpretability. While addressing these problems, neuro-symbolic training introduced scalability and training issues which limited its effectiveness to overly simplistic domains. In the age of foundation models, where prompting alone is enough to solve many tasks without training, we highlight three pitfalls of neuro-symbolic training with respect to compute, data, and programs. These pitfalls are avoided by neuro-symbolic prompting which replaces training with prompting of foundation models to offer the benefits of neuro-symbolic without the downsides of training. Finally, we encourage future research on neuro-symbolic prompting systems which infer the necessary symbols and program for solving a problem, instead of requiring them to be known in advance.

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

## A  Prompts

The prompt we use for the foundation models for all benchmarks takes the following form with placeholders that depend on the particular dataset.

```
Prompt

System Prompt:
You are a helpful assistant.

User Prompt:
After examining the input, determine <output_description>. Here are some ↩
    examples:
Example 1:<ex1_input>This is an example of <ex1_output>.
...
<input>The input is <input_description>. Examine it and then output just <↩
    output_description> after 'FINAL ANSWER:'. If unsure of the answer, try to ↩
    choose the best option.

Assistant:
```

For Sum5 the input description is "an image of a handwritten digit", the output description is "the digit as an integer from 0 to 9", and we use 5 few-shot examples.

For HWF5 the input description is "a handwritten number from 0 to 9", the output description is "the value of the number as an integer from 0 to 9", and we use 5 few-shot examples for the digit perception. For operator extraction the input description is "a handwritten arithmetic operator", the output description is "the operator as a string in the set '+', '-', '*', '/' (note that the division operator can look like a line with a dot above and below it and multiplication can look like an 'x')" and we use 4 few-shot examples.

For CLUTRR the input description is "a description of a relationship between two people and a query about the two people's relationship", the output description is

```
the described relationship which answers the question. Use the
    ↪ pronouns to determine the people's gender. The relationship
    ↪ must be one of the following: {'brother', 'sister', 'father', '
    ↪ mother', 'son', 'daughter', 'grandfather', 'grandmother', '
    ↪ uncle', 'aunt', 'nephew', 'niece', 'husband', 'wife', 'brother-
    ↪ in-law', 'sister-in-law', 'son-in-law', 'daughter-in-law', '
    ↪ father-in-law', 'mother-in-law', 'grandson', 'granddaughter', '
    ↪ unknown'}. For example, for the input 'John took his sister
    ↪ Mary to the store. John is Mary's what?' the output should be '
    ↪ brother.' Output just the relationship as a word.
```

and we use 2 few-shot examples.

For CLEVR the input description is "an image of geometric objects", the output description is

```
each object's bounding box and attributes in the form {\"bbox_2d\": (
    ↪ x1, y1, x2, y2), \"attributes\": (color, shape, material, size)
    ↪ \}. Colors can be one of ['gray','green','blue','red','brown','
    ↪ purple','yellow','cyan'], shapes can be one of ['cube','
    ↪ cylinder','sphere'], material can be one of ['rubber','metal']
    ↪ (it is rubber if the finish is matte and metal if shiny), and
    ↪ size can be one of ['small','large'].
```

and we use 2 few-shot examples.

For Leaf, we use three different prompts for the three networks used for perception in the neuro-symbolic training program. For all networks, the input description is "an image of a leaf". For the margin network, the output description is "the classification of the leaf's margin as one of 'entire', 'indented', 'lobed', 'serrate', 'serrulate', 'undulate'", and we use 5 few-shot examples. For the shape

network, the output description is "the leaf's shape as one of 'elliptical', 'lanceolate', 'oblong', 'obovate', 'ovate'" and we use 9 few-shot examples. Finally, the output description for the texture network is "the classification of the leaf's texture as one of 'glossy', 'leathery', 'smooth', 'rough'" and we use 3 few-shot examples.

# B  Experiment Details

For all prompting experiments, we use greedy decoding (temperature 0) so there are no error bars for neuro-symbolic prompting methods.

## B.1  Setup

We describe the benchmark datasets, Foundation Models, and NeSy learning baseline below.

**Datasets**  We use five standard NeSy benchmarks:

- Sum5 [11]: Constructed from the MNIST dataset of handwritten digits [22]. The input consists of five images of digits and the expected output is the sum of the digit values.
- HWF5 [23]: This dataset consists of five images creating an arithmetic expression. There are three handwritten digits from zero through nine and two handwritten operators representing addition, subtraction, division, and multiplication. The expected output is the evaluation of the expression.
- CLUTRR [24]: The input consists of natural language paragraphs describing family relationships and a question about the relationship between two people mentioned.
- CLEVR [26]: The input is an image containing various objects of different shape, size, color, and texture along with a question about the image.
- Leaf [12, 25]: The input is an image of a leaf and the expected output is the species of the leaf.

**Models**  We evaluate Foundation Model prompting as a replacement for neural network training in NeSy learning using the following Foundation Models:

- Phi-3.5 Vision Instruct [30]
- Qwen2.5 VL Instruct (3B, 7B, and 72B) [28]
- InternVL 2.5 Instruct (8B, 38B, and 78B) [29]
- Llama 3.2 Vision Instruct (11B and 90B) [27]
- Gemini 2.0 Flash [31]

**NeSy learning baselines**

- Scallop [11]: We use Scallop as a representative NeSy learning method.
- ISED [12].

## B.2  Full NeSy Learning vs. Foundation Model Prompting Performance Gap

The performance gap between NeSy prompting and full NeSy learning is quickly diminishing. In addition, the performance gap reduces with increasing model scale. This is shown in Figure 2. Results labelled with "—" for ISED are due to an unavailable implementation for the dataset. For GPT-4o, we only evaluate on two datasets to reduce cost. Finally, the neuro-symbolic prompting results marked "—" for the CLEVR dataset are due to those models not supporting object bounding box generation.

# C  Human Evaluation For Leaf

Since the Leaf dataset does not provide ground truth annotations for the three leaf properties, we perform a human evaluation to quantify symbol hallucination in Scallop. For the study, we took all problems that were correctly answered by Scallop while incorrectly answered by neuro-symbolic

Table B.1: All results

| Method | Sum5 | HWF5 | CLUTRR | CLEVR | Leaf |
|---|---|---|---|---|---|
| Scallop | $0.975 \pm 0.002$ | $0.966 \pm 0.005$ | $0.400 \pm 0.031$ | 0.750 | $0.811 \pm 0.035$ |
| ISED | $0.923 \pm 0.004$ | 0.023 | — | — | $0.823 \pm 0.041$ |
| Phi-3.5-vision-instruct | 0.17 | 0.01 | 0.53 | — | 0.055 |
| Llama-3.2-11B-Vision-Instruct | 0.645 | 0.0 | 0.285 | — | 0.255 |
| Llama-3.2-90B-Vision-Instruct | 0.655 | 0.180 | 0.626 | — | 0.178 |
| Qwen2.5-VL-3B-Instruct | 0.075 | 0.015 | 0.560 | 0.250 | 0.215 |
| Qwen2.5-VL-7B-Instruct | 0.595 | 0.03 | 0.640 | 0.650 | 0.335 |
| Qwen2.5-VL-72B-Instruct | 0.790 | 0.250 | 0.790 | 0.900 | 0.390 |
| InternVL2.5 8B | 0.540 | 0.025 | 0.150 | 0.160 | 0.250 |
| InternVL2.5 38B | 0.825 | 0.140 | 0.730 | 0.730 | 0.335 |
| InternVL2.5 78B MPO | 0.830 | 0.000 | 0.760 | 0.880 | 0.405 |
| GPT-4o | 0.860 | — | — | — | 0.509 |
| Gemini-2.0-Flash | 0.815 | 0.710 | 0.760 | 0.765 | 0.405 |

Q20

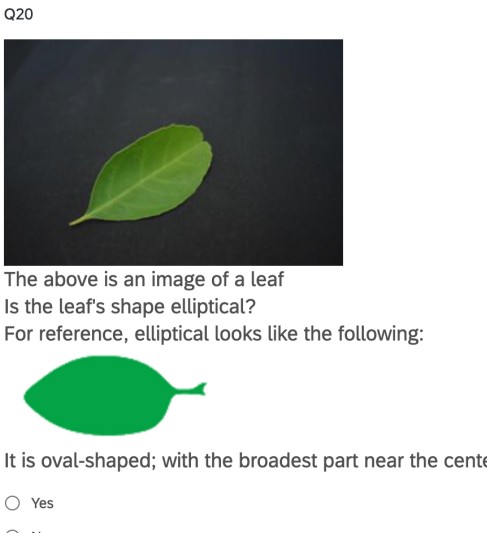

The above is an image of a leaf
Is the leaf's shape elliptical?
For reference, elliptical looks like the following:

It is oval-shaped; with the broadest part near the center.

○ Yes

○ No

Figure C.1: Example question asked in the Leaf dataset human evaluation. Every question was a binary choice where the choice 'Yes' always corresponded to agreeing with Scallop and 'No' corresponded to agreeing with Gemini, although this was not told to participants.

prompting with Gemini, sampled 15 questions for each of margin, shape, and texture which had differing property predictions for the two methods. We then directly asked participants if the neural model's prediction from Scallop was correct for each of the three properties, consisting of a total of 45 questions (15 for each property). For each property we also provide a description and small depiction to help participants accurately classify the leaves. An example question for the subset about the leaf shape is in Figure C.1.

In total, we hired 10 people through Prolific [34] to perform the evaluation, answering all 45 questions each. Respondents were required to have at least a high school education and speak English as their first language. We paid an average of $16.76 per hour and the task took an average of 6 minutes and 16 seconds.

# D   Additional Results

To further support the findings in the main paper, we provide additional results for each of the pitfalls from the main paper.

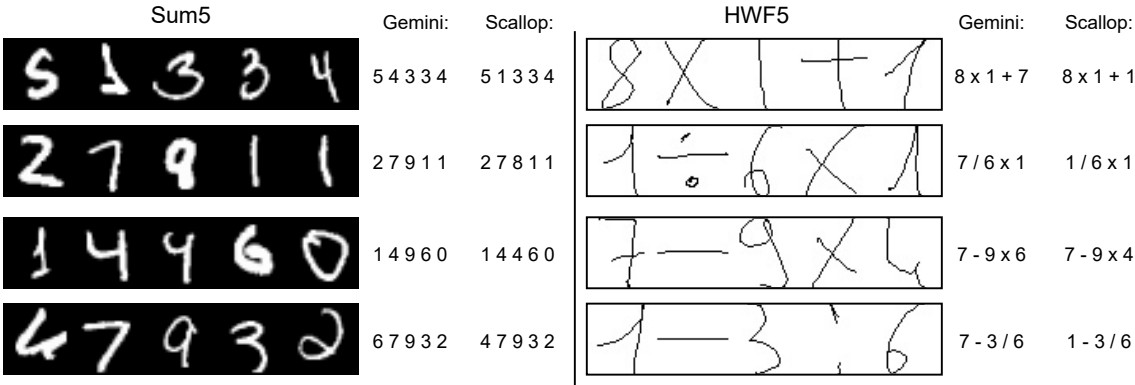

Figure D.2: Additional examples of the data pitfall in Sum5 and HWF5 datasets.

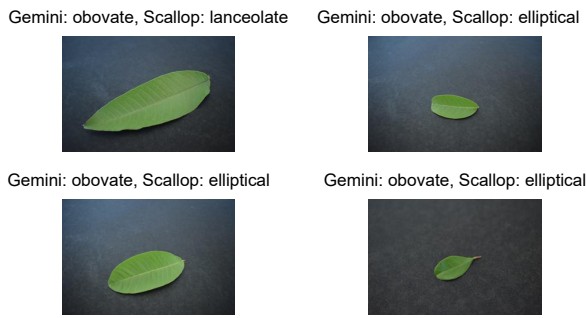

Figure D.3: Additional examples of the data pitfall in the Leaf dataset.

## D.1 The Data Pitfall

Additional qualitative results are included in Figure D.2.

## D.2 The Program Pitfall

Additional qualitative results are included in Figure D.4.

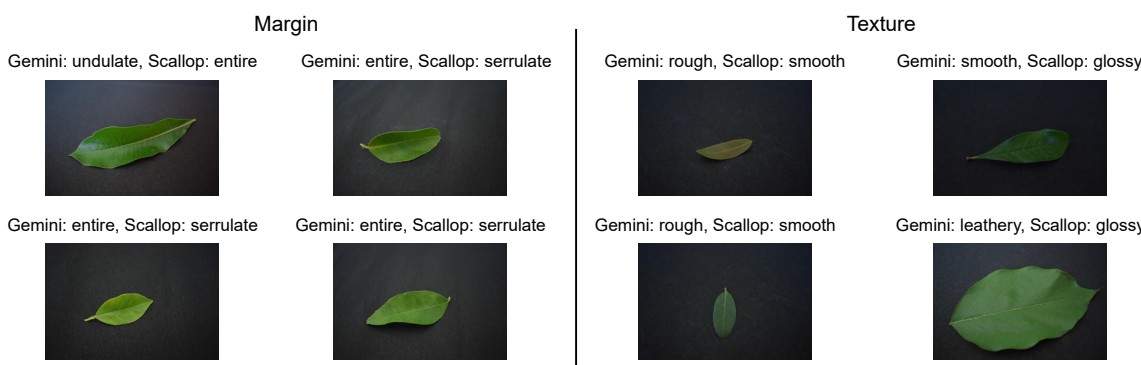

Figure D.4: Additional examples of the program pitfall in the Leaf dataset.

