# OpenReview forum: "The Road to Generalizable Neuro-Symbolic Learning Should be Paved with Foundation Models"
_NeurIPS.cc/2025/Position_Paper_Track — Submitted to NeurIPS 2025 Position Paper Track_

### Official Review · Reviewer_inQr · 2025-08-04

**Significance:** 2
**Presentation:** 4
**Rating:** 3
**Confidence:** 3

**Summary:**

The paper argues that with the advent of foundation models neuro-symbolic training should be substituted by neuro-symbolic prompting.
The argument is supported by two main observations
- Computation level: it is not necessary to train from scratch as current foundation models are already good at extracting good symbolic representations when properly prompted on traditional neuro-symbolic benchmarks.
- Representation level: neuro-symbolic training is based merely on a top-down approach resulting in the symbol grounding problem (e.g. the neural network learns the wrong input-output mapping)
Both observations are backed up by experimental analysis.

**Strengths:**

- The paper is very clearly written and I enjoyed the reading **Clarity**
- The hypothesis questions and the experimental analysis are very original and compelling to support the proposed approach **Originality**.
- The work may be of interest to the neuro-symbolic community **Scope**.


**Weaknesses**
- The paper seems to propose a solution rather than a position **Significance**. As a matter of fact, clear challenges and a roadmap are currently missing and should be explicitly provided for a position paper.
- It is not clear what is the difference between the proposed solution and constrained decoding used in LLMs for constrained generation **Significance**
- There is an important misconception, which is an artifact of the naming conventions. Note that neuro-symbolic training differs from the problem of neuro-symbolic integration. One can achieve the former through a simple top-down approach, whereas the latter requires a two-way communication between the neural and symbolic components. The paper focuses on the former and therefore has limited potential impact **Significance**

**Weaknesses:**

- see above *Strengths*
- Imprecise related work. The symbol grounding problem was first introduced by [1] in [2]. The work in [3] first revived the simple grounding problem in the machine learning community through a simple instantiation using a sequence of MNIST images. Later Marconato et al. [4] extended this idea to a larger set of tasks, giving a different name to the same problem. Again reasoning shortcuts is a misnomer as the problem arises simply from the fact that learning occurs in a top down fashion, where the neural network is informed only by the supervisory information of the program, thus not necessarily capturing information of the input (indeed symbols should be grounded on perceptual information, hence the name symbol grounding problem). The recent work in [5] demonstrates that a joint combination of bottom up and top down training is sufficient to mitigate this issue.
- continued in *Questions*

**Questions:**

- The discussion about program and data pitfalls seems to be an effect of the symbol grounding problem **Discussion**.
    - Program pitfalls “using programs as a component in neuro-symbolic training can lead to the neural component hallucinating symbols” -> this is the problem of symbol grounding as neuro-symbolic training is typically performed in a top down fashion
    - Data pitfalls “neuro-symbolic training with specialized datasets, as opposed to large-scale foundation model pretraining, encourages overfitting to dataset particularities.” -> This is again a problem of top-down training

**References** \
[1] The symbol grounding problem. Physica D: Nonlinear Phenomena 1990 \
[2] Perceptual symbol systems. Behavioral and Brain Sciences 1999 \
[3] Learning Symbolic Representations Through Joint GEnerative and DIscriminative Training. ICLR Workshop 2023 \
[4] Neuro-symbolic continual learning: Knowledge, reasoning shortcuts and concept rehearsal. ICML 2023 \
[5] Unifying Self-Supervised Clustering and Energy-Based Models. To appear in TMLR 2025

**Alternative Position:**

No

**Author Identification:**

No.

**Context:**

2

**Details Of Ethics Concerns:**

-

**Discussion:**

3

**Ethics:**

["NO or VERY MINOR ethics concerns only"]

**Position:**

No, the paper argues that a specific technical approach is superior to other approaches.

**Support:**

3

**Thoroughness:**

4

---

### Official Review · Reviewer_XE9f · 2025-08-12

**Significance:** 3
**Presentation:** 3
**Rating:** 5
**Confidence:** 4

**Summary:**

This paper claims that the traditional neuro-symbolic training introduced scalability and training issues which limited its effectiveness to overly simplistic domains. The authors believe that prompting along is often enough to solve many tasks without training in the age of foundational models. They highlight three pitfalls of neuro-symbolic training with respect to compute, data and programs, and encourage future research on neuro-symbolic prompting systems which infer the necessary symbols and program for solving a problem.

**Strengths:**

1. The paper provides clear arguments with empirical evidence on various datasets for its position.
2. The proposal of the paper may be interested to the neuro-symbolic learning community.

**Weaknesses:**

1. The paper doesn't address the pitfalls of traditional neuro-symbolic learning by leveraging data augmentation or synthetic data generation.

**Questions:**

Can you answer the questions in the weaknesses?

**Alternative Position:**

Yes, and alternative positions are well-considered and addressed by the argument

**Author Identification:**

No.

**Context:**

3

**Discussion:**

3

**Ethics:**

["NO or VERY MINOR ethics concerns only"]

**Position:**

Yes, the paper argues for or against a position related to machine learning.

**Support:**

3

**Thoroughness:**

4

---

### Official Review · Reviewer_M9N8 · 2025-08-13

**Significance:** 4
**Presentation:** 4
**Rating:** 8
**Confidence:** 3

**Summary:**

This position paper argues that neuro-symbolic prompting, a paradigm in which foundation models (FMs) are used for perception tasks like symbol extraction, while symbolic programs handle reasoning, offers a more scalable and generalizable alternative to traditional neuro-symbolic systems. In contrast, the traditional approach relies on end-to-end training of neural and symbolic components, which the authors argue is less effective in the era of foundation models. The authors identify three major pitfalls in the traditional approach: the compute pitfall (inefficient and costly training), the data pitfall (overfitting and lack of robustness), and the program pitfall (symbol hallucination due to weak supervision). Through empirical evaluations across five benchmarks, the paper demonstrates that prompting-based systems can match or outperform training-based ones, while offering improved interpretability and reducing the need for annotated data. The authors introduce the symbol hallucination rate as a novel metric to assess the reliability of intermediate representations and highlight autonomous program and symbol inference as a key direction for future research.

**Strengths:**

1. The paper presents a clear, well-structured argument for shifting from neuro-symbolic training to prompting, with well-defined pitfalls supported by both quantitative and qualitative evidence.
2. It is well-written and accessible to both neuro-symbolic and foundation model researchers, with strong diagrams, terminology, and a helpful taxonomy that guides critique and future work.
3. The topic is timely and relevant, addressing core concerns like interpretability, reliability, and efficiency in the context of evolving ML paradigms.
4. Extensive experiments across diverse benchmarks (Sum5, HWF5, CLUTRR, CLEVR, Leaf) compare traditional systems (Scallop, ISED) with prompting-based approaches using state-of-the-art FMs (e.g., Gemini, GPT-4o, LLaMA 3.2), grounding the argument in empirical evidence.
5. The paper introduces the novel symbol hallucination rate metric, offering a thoughtful way to assess the reliability of intermediate representations—validated through human evaluation.
6. It encourages a shift in research focus toward program synthesis and symbol inference, rather than training perception models from scratch.
7. Code is promised and experiments are replicable, enhancing the paper’s practical value.

**Weaknesses:**

1. The paper briefly mentions finetuning but largely dismisses it in favor of prompting. A more balanced discussion of its advantages especially in low-resource or domain-specific contexts would strengthen the argument.
2. The paper argues prompting is preferable because it avoids training. However, it overlooks the high inference costs of foundation models. Quantitative comparisons (e.g., compute time, energy, API costs) are missing and would be valuable for practitioners.
3. The proposed metrics, like symbol hallucination, rely on ground-truth intermediate symbols, limiting applicability to real-world, unannotated data. Practical evaluation strategies or proxy metrics are not provided.
4. Reliance on external, often closed-source foundation models raises productization concerns where model behavior may change across time which can break deterministic programs relying on specific FM outputs. The paper lacks discussion on versioning, regression testing, and safeguards against model drift.

**Questions:**

1. How well would neuro-symbolic prompting generalize to domains with limited or noisy data, such as medical imaging or legal reasoning?
2. How does the cost and latency of prompting large foundation models compare to training smaller, task-specific models in real-world deployment scenarios?
3. What evaluation protocols or proxy metrics would you propose when annotated symbols aren’t available?
4. In which regimes would training specialized neural components remain necessary, and how might hybrid fine-tuning + prompting approaches fit?

**Alternative Position:**

Yes, and alternative positions are well-considered and addressed by the argument

**Author Identification:**

No.

**Context:**

4

**Discussion:**

4

**Ethics:**

["NO or VERY MINOR ethics concerns only"]

**Position:**

Yes, the paper argues for or against a position related to machine learning.

**Support:**

4

**Thoroughness:**

3

---

### Note · Authors · 2025-09-04

**1-11 Submit Again:**

Definitely yes

**1-1 Submission Process:**

5

**1-2 Next Year:**

I would like if the position paper track had a traditional rebuttal process with an author-reviewer discussion period. The current system makes it unclear how to respond directly to reviewers and even what the purpose of the survey is.

**1-4 Interest:**

["Panel discussions with other position paper authors", "Structured debates on controversial topics"]

**1-5 Thoughtful:**

6

**1-6 Supportive:**

6

**1-7 Technical Aspects Versus Position:**

8

**1-8 Gate Keeping:**

3

**1-9 Camera Ready Changes:**

- Cost factors: we will add a discussion of inference and training costs in Section 2.1.
- Model drift safeguards: we will add a discussion of model drift and safeguards such as versioning and regression testing to the Alternate Views section.
- Terminology and related work: we will clarify that “neuro-symbolic training” is different from “neuro-symbolic integration” in related work, but that we adopt a simple definition of neuro-symbolic training (as shown in Figure 1) to avoid any terminology confusion. We will also update our related work section to add five papers related to the symbol grounding problem.
- Symbol grounding problem: we will explicitly link our program and data pitfalls to the symbol grounding problem in the related work section, and we will clarify that foundation models are trained in a bottom-up approach which can mitigate some of the symbol grounding issues of top-down trained models.

**3-1 Review Response1:**

M9N8

**3-2 Reaction To Review1:**

We thank the reviewer for their thoughtful comments and strong support of our paper. This review is highly thoughtful, supportive of our position, and engaged with the position itself by asking several relevant follow up questions. We feel no gatekeeping from the reviewer. We respond to the noted weaknesses and questions in detail below.

**W1 / Q1 & Q4.** We discuss finetuning in the Alternate Views section, noting it can be advantageous in low-resource domains. We focused on prompting to clearly position it against specialized neuro-symbolic training. As we stated on lines 277-283 of the paper, we view finetuning as a middle ground between neuro-symbolic prompting and neuro-symbolic training which is still preferable over neuro-symbolic training. Even in traditionally low-resource areas like medicine, foundation models are becoming highly knowledgeable, yet we agree that finetuning alongside prompting can be necessary for strong performance.

**W2 / Q2.** We agree that the inference costs of foundation models are higher than small specialized models, but they provide higher overall accuracy, stronger generalization, and reduced issues such as symbol hallucination. In addition, the training of specialized models requires annotated data which is often itself expensive to obtain. We will include a discussion of these cost factors in Section 2.1 (the compute pitfall) of the revision.

**W3 / Q3.** Evaluating symbol hallucination without ground truth symbols is challenging. In the paper, we evaluated symbol hallucination on the Leaf dataset which lacked ground truth symbol labels by using crowd workers to manually label our data. While this method is not scalable, it can be used for evaluating a subset of one’s data to get an estimate of the symbol hallucination rate.

**W4.** Following the reviewer’s suggestion, we will include a discussion of model drift and safeguards such as model versioning and regression testing in the Alternate Views section.

**3-3 Review Response2:**

XE9f

**3-4 Reaction To Review2:**

We thank the reviewer for their review. We find this review is not very thoughtful as it brings up only a single weakness regarding data augmentation which is not very relevant to our position. The reviewer does not seem to take a side in supporting our position, but it is focussed on the position itself rather than on technical details of the paper. We do not feel any gatekeeping from the review. We respond to the reviewer’s comment in detail below.

**Regarding data augmentation and synthetic data generation**
The three pitfalls of neuro-symbolic training still remain even if one uses data augmentation or synthetic data generation to mitigate some of the problems. The main benefit of data augmentation is in improving the generalization of trained models, so the compute pitfall and program pitfalls are unaffected since the compute pitfall concerns in-distribution performance and the program pitfall is about using a single program for neuro-symbolic training. Data augmentation can mitigate some of the issues of the data pitfall, but this depends on choosing the right augmentations, and augmentations often are not general enough. For instance, it is unclear what data augmentations would help the neural network not memorize the peculiarities of the Leaf dataset as shown in Figure 3.

**3-5 Review Response3:**

inQr

**3-6 Reaction To Review3:**

We thank the reviewer for their thoughtful and engaged comments. The review mainly requests clearer terminology and links to prior work, which we will address. Below we respond to the key points.

**Our position** Our position is stated in the title, abstract, and throughout the paper: *foundation models enable generalizable neuro-symbolic solutions, offering a path to the original goals of neuro-symbolic learning without the downsides of training from scratch.* We analyze three pitfalls of end-to-end neuro-symbolic training (program, data, model) and argue prompting avoids them. In *Looking Ahead*, we outline a roadmap calling for inferring both symbols and programs.

**Constrained decoding** Prompting is orthogonal to constrained decoding. Constrained decoding enforces structure at decode time but does not provide symbolic reasoning or steer content. In our experiments, foundation models already followed format instructions, so constrained decoding was unnecessary.

**Neuro-symbolic integration** By neuro-symbolic training we mean systems trained end-to-end with final-label supervision, where $y = M(P(x))$ with $M$ neural and $P$ symbolic, and gradients must pass through $P$. We do **not** restrict this to top-down regimes. We compare against “neuro-symbolic integration” methods trained with Scallop. For example, in Scallop Sum5 the forward pass symbolically sums predicted digits, and the backward pass differentiates through that addition.

**Related work** We thank the reviewer for pointing us to relevant symbol-grounding papers. We will add them and refine our treatment of reasoning shortcuts.

**Pitfalls and symbol grounding** We agree these pitfalls manifest as grounding failures. They persist under top-down training due to small datasets (data pitfall) and brittle programs (program pitfall). In contrast, foundation models pretrained bottom-up on large-scale perceptual data largely avoid these issues. We will make this link explicit in revision.

---

### Meta-Review · Area_Chair_g7a8 · 2025-09-12

**Rating:** 7
**Confidence:** 4

**Strengths:**

The paper presents a clear, well-structured argument. and provide sufficient evidence. The position is relevant to the NeurIPS community, and I would like to see further discussion at the conference.

**Weaknesses:**

no

**Questions:**

see reviewers' questions

**Ethics:**

no ethical issues

**Thoroughness:**

3

---

### Decision · Program_Chairs · 2025-09-26

Reject